# Effect of Ni on the Microstructure and Diffusion Behavior at the Interface of WC/High-Speed Steel Composites

**Hongnan Li [1], Ling Yan [2,*], Linghao Meng [1,3], Yan Li [2], Fangfang Ai [2], Hongmei Zhang [1,2,*] and Zhengyi Jiang [4]**

[1] School of Material and Metallurgy, University of Science and Technology Liaoning, Anshan 114051, China; lihongnan7270@ustl.edu.cn (H.L.); womenglingh@163.com (L.M.)

[2] State Key Laboratory of Metal Material for Marine Equipment and Application, Anshan 114009, China; 2323liyan@sina.com (Y.L.); aifangfang@163.com (F.A.)

[3] Beijing Shougang Co., Ltd. Marketing Center, Beijing 100041, China

[4] School of Mechanical, Materials, Mechatronic and Biomedical Engineering, University of Wollongong, Wollongong, NSW 2522, Australia; jiang@uow.edu.au

\* Correspondence: yanling_1101@126.com (L.Y.); zhanghm@ustl.edu.cn (H.Z.); Tel.: +86-139-4122-0530 (L.Y.); 86-138-0492-7151 (H.Z.)

**Abstract:** In this study, WC-Ni/high-speed steel composite materials for application as micro-drills were prepared by cold pressing and high-temperature vacuum sintering using a self-designed mold in the laboratory. The effect of Ni on the microstructure and diffusion behavior at the interface of the WC/high-speed steel composite was investigated. The results show that the addition of Ni promoted the diffusion of elements, and reduced defects such as micropores and microcracks at the WC/high-speed steel composite interface. It also improved the bonding strength of the WC/high-speed steel composite interface, and significantly decreased the WC hardness.

**Keywords:** Ni element; WC/high-speed steel composite; microstructure; microhardness; interface diffusion behavior

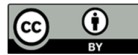

## 1. Introduction

With advancements in the electronics industry, there is an increasing demand for light, thin, and miniaturized fine components for use in electronic products. The design of high-performance micro-drills for use in process printed circuit boards has also been increasing recently. Tungsten carbide (WC) materials have been widely used for nuclear fusion devices and medical equipment, and in the mining industry, because of their desirable properties, including high hardness, high melting point, high density, and high wear resistance [1–5]. Currently, WC is a widely used material for micro-drills. WC micro-drills provide excellent performance in terms of hardness and wear-resistance. Due to the poor fracture toughness and high price of WC, its applications have been limited. Parts composed of WC may fail due to brittle behavior, especially in applications such as precision micro-drilling. In the process of WC sintering, the binder needs to have a certain degree of wettability to the hard phase and a relatively low melting point. Generally, Co is widely used as a binder in WC materials. WC-Co has high grindability and high hardness, and is widely used in the fields of cutting tools, molds, and machinery [6]. WC and Co are considered to be well-matched in terms of their mechanical properties and manufacturing processes, with good wettability, high hardness, and good toughness [7,8]. However, the scarcity of Co resources and the radiative decay of Co can cause nuclear pollution, and Co is not suitable in corrosive environments, particularly in acidic media. In addition, its toxicity, rising prices, and the geopolitical issues involved in primary mining have prompted people to research and develop low-cobalt binders, preferably cobalt-free binders [2–9]. The traditional cemented carbide materials with metal Co as the binder phase have the advantages of high strength and toughness, but their hardness, resistance

grindability, and chemical stability are insufficient [10,11]. Fernandez et al. [12,13] considered the scarcity of Co resources and the fracture toughness of cemented carbides. They developed a series of stainless-steel powders as substitutes for cobalt in cemented carbides. The hardness and fracture toughness of the obtained samples are obviously improved, but the cost problem is not solved to a great extent.

The global reserves of Ni are 70 times those of Co, so Ni is a relatively cheap metal that does not pollute the environment. Ni and Co are adjacent in the periodic table and have similar physical and chemical properties, so Ni can often be used as a good substitute for Co. The earliest research on WC–Ni alloys was reported by German and former Soviet Union researchers in the 1930s [14]. Due to the uneven distribution of WC and Ni after high-temperature sintering, the Ni pool formed by Ni aggregation and abnormal growth of WC grains readily occurs in the WC–Ni microstructure, which reduces the strength and toughness of WC–Ni alloys [15]. To obtain higher strength, hardness, corrosion resistance, and grindability, rare earth and grain growth inhibitor have been added to WC–Ni alloys. Recently, WC–Ni cemented carbide was developed, which has high hardness, strength, and thermal stability, as well as excellent toughness and resistance grindability [16–20].

Yong et al. [21] synthesized WC–Co nanocomposites using a new method to prepare nanocrystalline composites; the prepared nanocomposites had excellent comprehensive properties. Ghosh et al. [22] studied the static and dynamic wetting behavior of hard WC–Co coatings. Krishna et al. [23] used WC and Co mixed particles as the reinforcement in the Al7075 (Dongguan Jiajin Metal Co., Ltd, Dongguan, China) matrix to prepare composite materials by liquid metallurgy. The results showed that the tensile strength of the composite material was significantly improved. Liu et al. [24] used ultrafine WC and Ni powder as the main raw materials, adjusted the total carbon content in the mixture, and obtained WC–15Ni high-performance cemented carbide through vacuum sintering. Jia et al. [25] used laser cladding technology to repair damaged parts with WC/Ni powder. The results show that the repair position has better wear resistance and hardness. Zhao et al. [26] used laser remelting technology to treat the flame-sprayed, Fe-based Ni/WC cermet coating. The results show that there are holes and interlaminar cracks on the coating interface, which have typical mechanical bonding characteristics. Wang et al. [27] introduced ultrasonic waves into the WC–Ni laser melting pool through air to form a cladding coating on the surface of 316L steel. The WC–Ni coating was prepared by ultrasonic vibration to obtain better grindability.

The hard phase plays a key role in the hardness and grindability of the material, and the bonding phase has important impacts on the strength and toughness of traditional cemented carbide. In general, alloys with higher contents of binder phase have higher toughness and lower hardness, and alloys with higher hard phase contents have higher hardness and lower toughness. Therefore, simultaneously improving the hardness and toughness of cemented carbide to prepare double high (high hardness and high toughness) cemented carbide has become an important topic in the field of cemented carbide. To solve the contradiction between toughness and hardness of traditional carbide tool materials, the following methods have been proposed: reducing the size of powder particles and preparing a nanometer WC matrix composite [28–30]. The gradient structure was introduced to prepare gradient WC matrix composites to overcome the shortcomings of materials with a uniform structure, and introduce changes in the microstructure or composition in the materials, so that the material will have specific functionality at a specific position. As such, overall, the material has significantly better properties than homogeneous materials [31,32].

Micro-drills were originally composed of monolithic cemented carbide. To reduce costs, the structure of micro-drills has changed from the whole cemented carbide micro-drill to a stainless-steel handle and cemented carbide butt welding micro-drill, and then to the latest type, the non-equidiameter plane joint-welding between stainless steel shank and carbide micro-drill. However, these drills still suffer from the problem of bit toughness [33]. Generally, the four kinds of molybdenum-based high-speed steel (HSS, Kobe

Steel Corporation, Kobe, Japan) used in industry are M1, M2, M7, and M10 models. In the past, M2 was mainly used to make twist bits. Today, the proportion of M2 in twist bits is declining due to the need to drill various materials with greater grindability; however, small-diameter bits are often made with M2, which has higher toughness [34]. To meet the increasing production and performance requirements of the bit material industry, it is necessary to develop a composite material with excellent performance. Advanced micro-forming technology can reduce the cost of manufacturing cemented carbide and result in a composite material with high hardness and high toughness [35].

To produce micro-drill materials with excellent comprehensive performance with a low-cost and simple process, we adopted a direct powder–solid bonding method, with HSS used as the inner core and WC particles used as the outer sleeve, that is, manufactured to realize powder–solid diffusion bonding between WC powder and HSS under the dual action of cold pressure and high temperature, without the use of an intermediate layer. Nanometer WC particles and Ni powders were used to modify the strength and toughness of the materials. WC–Ni was prepared using a powder metallurgy (P/M) process. The solidification temperature of WC–Ni ultrafine powders is much lower than that of the theoretical melting point [36], which simplifies the manufacturing process.

## 2. Materials and Methods

The experimental materials included WC powder with a particle size of 200 nm and Ni powder with a particle size of 0.5–2.0 μm. The added amount of Ni powder was 10% of the quality mixed powder. WC and Ni powders were mixed in a silicon nitride ball mill at a speed of 200 rpm for 1 h using a planetary ball mill (Retsch GmbH, Haan, Germany). The mixed powders and M2 HSS with a diameter of 1 mm were put into the mold (Figure 1), to construct a preliminary micro-drill with a diameter of 3 mm.

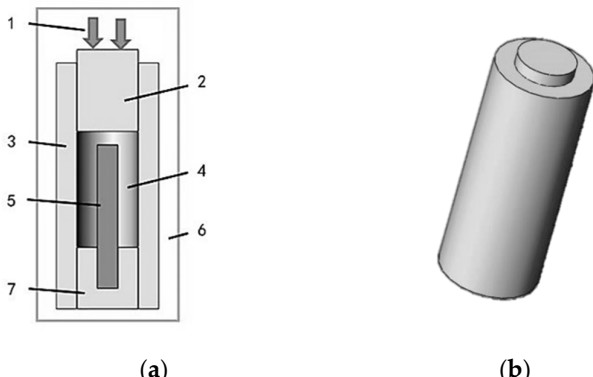

(**a**)                    (**b**)

**Figure 1.** Tungsten carbide (WC)/high-speed steel composite preparation mold. (**a**) Parts of the mold are: 1, pressure; 2, upper punch; 3, pressing die body; 4, mixed material powder; 5, high-speed steel core; 6, shielding gas; 7, stationary punch. (**b**) Three-dimensional mold diagram.

In the experiment, WC powder and a high-speed steel core were pressed in the mold using a universal testing machine (Jinan Chenda Testing Machine Manufacturing Co., Ltd., WDW-300, Zhong Lu Chang, Jinan, China). Because the composite material was sintered at high temperature, 310s heat-resistant steel was selected as the die material. The compression process was performed at room temperature with a compression speed of 1 mm/min and a maximum pressure of 1145 MPa. The pressed samples were sintered in a vacuum tube furnace (GR.TF60, Shguier, Shanghai, China) at a rate of 8 °C/min to 1280, 1300, and 1320 °C. The holding time was 90 min, and then the samples were cooled to room temperature in a furnace. The specimens were inlaid at 170 °C for 20 min with a mosaic machine (XQ-2B, CEWEIGUANDIAN, Beijing, China).

The microstructure was observed with a digital microscope (VHX-5000, Keyence (China) Co., Ltd., Shanghai, China). The microstructure and interface of the composite were observed using SEM (SEM Zeiss-IGMA HD, Jena, Germany). The element distribution and diffusion at the interface of the composite were observed by energy-dispersive spectrometry (EDS Zeiss-IGMA HD, Jena, Germany). An X-ray diffractometer (X'Pert Powder, PANalytical, Beijing, China) was used to analyze the phase at the interface of the composite materials. The Vickers hardness of the cross-section of the WC/HSS composites was tested using a microhardness tester under the conditions of loading weight of 0.1 kg and holding time of 10s (Q10M, Qness GmbH, Salzburg, Austria). In this experiment, to study the changes in the microstructure of the WC/HSS and WC–Ni/HSS composites at different sintering temperatures, a VL2000DX ultra-high-temperature laser confocal microscope (Yonekura MFG Co., LTD, Osaka, Japan) was used to observe the prepared samples (Argon atmosphere). The porosity was detected using Image J software (National Institutes of Health, Maryland, USA) [37,38].

## 3. Results and Analysis

### 3.1. Effect of Adding Ni on the WC Microstructure of WC/HSS Composites

3.1.1. Effect of Adding Ni on the WC Sintered Porosity

Figure 2 shows the WC microstructure of the WC/HSS composite material with and without added Ni at a sintering temperature of 1320 °C for 90 min.

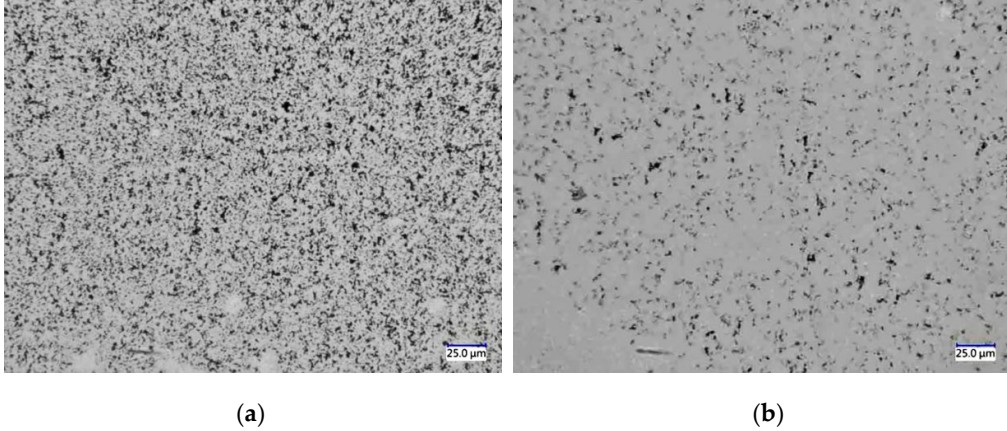

(**a**)　　　　　　　　　　　　　　　　　　　　(**b**)

**Figure 2.** The 1320 °C WC microstructure (OM): (**a**) no Ni added and (**b**) with Ni added.

Figure 2a presents the WC microstructure of the WC/HSS composite without added Ni. It can be seen that there is a large number of uniform micropores in the WC microstructure. The WC sintered density was 75.87%. The WC microstructure of the WC–Ni/HSS composite with added Ni is shown in Figure 2b. The pores in the WC structure were significantly reduced with the addition of Ni, and a large dense structure formed. The WC sintered density with added Ni was 94.16%, which was about 7% higher than the WC sintered density with added Co studied by Wang et al. [11].

From the results, it can be seen that the WC sintered density was improved by adding Ni to WC particles. As the melting point of Ni is lower than that of WC, Ni diffuses and distributes around WC particles as the sintering temperature is increased. With the increase in sintering temperature, the WC particles gradually grow in size, which compensates for the pores in the WC structure, thus the WC sintered density is enhanced after sintering.

### 3.1.2. Effect of Adding Ni on the Grains of the WC Microstructure

The effect of Ni addition on the microstructure and grain size distribution of WC/HSS under a sintering temperature of 1320 °C and holding time of 90 min is shown in Figure 3.

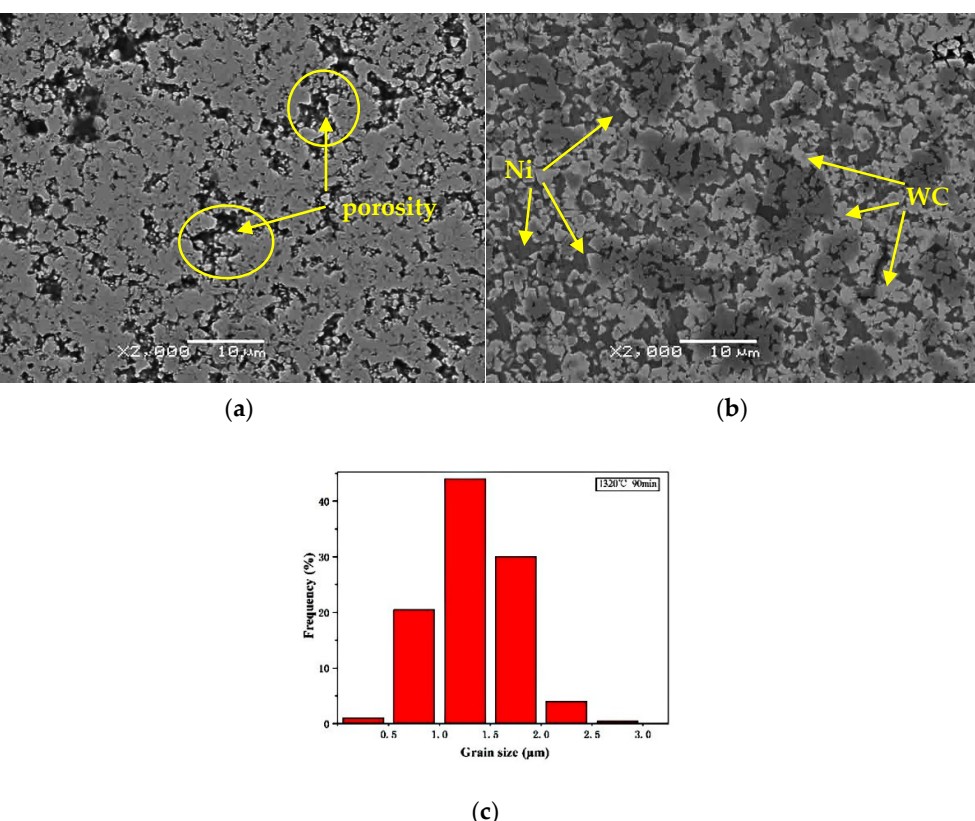

(**a**)                                                                          (**b**)

(**c**)

**Figure 3.** The 1320 °C WC microstructure (**a**) with no Ni added (**b**) with Ni added and (**c**) grain size distribution diagram with Ni added.

The WC microstructure of the WC/high-speed steel composite material with no Ni added is shown in Figure 3a. The defects of the WC structure can be clearly observed; the WC grains have a dense structure formed by large-area agglomeration, which led to the blurring of the grain boundaries. There is a large grain size, which belongs to the abnormal growth grain. A large number of grains belong to continuous normal growth. In this experiment, the size of the ultrafine WC powder was 200 nm, with a large specific surface area and high total surface energy of the grains. The powder easily aggregated, resulting in increases in the total grain boundary area and the interface energy. The continuous normal growth of grains mainly occurred during the heating and holding stages. Most grains grew uniformly almost simultaneously. The driving force of grain growth is mainly the decrease in the total interfacial energy, that is, the decrease in interfacial area. Figure 3b,c shows the WC microstructure of the WC–Ni/HSS composite with Ni added. It can be seen that the distribution of WC, Ni, and defects is shown, and the WC structure can be seen to be relatively compact, with only a few defective pores. Due to the existence of a binder phase, the morphology of WC grains was clear. There were many irregular polygon WC grains with an average grain size of about 1.33 μm. There was a large number of grains between 0.5 and 2 μm in size, indicating the continuous normal growth of grains. This may be due to Ni beginning to melt during sintering and diffusing into the pores between WC grains as the temperature was increased, which hindered the agglomeration of WC particles to a certain extent, reduced the total surface energy, and inhibited the

abnormal growth of WC grains. Because of the presence of Ni, the agglomeration between WC grains was hindered, meaning that less of the dense structure was formed and the WC content per unit area was reduced, which led to a decrease in the WC hardness.

*3.2. Influence of Ni Addition on the Interface of WC/HSS Composite Materials*

3.2.1. Influence of Ni Addition on the Interface Morphology of Composite Materials

A comparison of the interfaces of the WC/HSS composite material with no binder and with Ni binder at a sintering temperature of 1300 °C and a holding time of 90 min is shown in Figure 4. The upper half is the WC area, while the lower half is the high-speed steel area. Figure 4a presents the composite interface of the WC/HSS composite with no Ni added. There was a large number of pores and cracks in the WC area and large cracks at the bonding interface due to the high sintering temperature used, the different thermal expansion coefficient, the diffusion of elements between the composite materials, and the phase change within the diffusion zone. After heat preservation, the sample cracked during cooling under the influence of thermal stress at the junction of the diffusion zone and the high-speed steel. Figure 4b presents the composite interface of WC–Ni/HSS composites with Ni added. There were small pores in the WC area, no long cracks, fewer pores at the bonding interface, and the interface transition is uniform. Adding Ni to the WC particles strengthened the interface of the WC and high-speed steel composite material, reduced the pores at the interface, and improved the bonding strength of the composite interface.

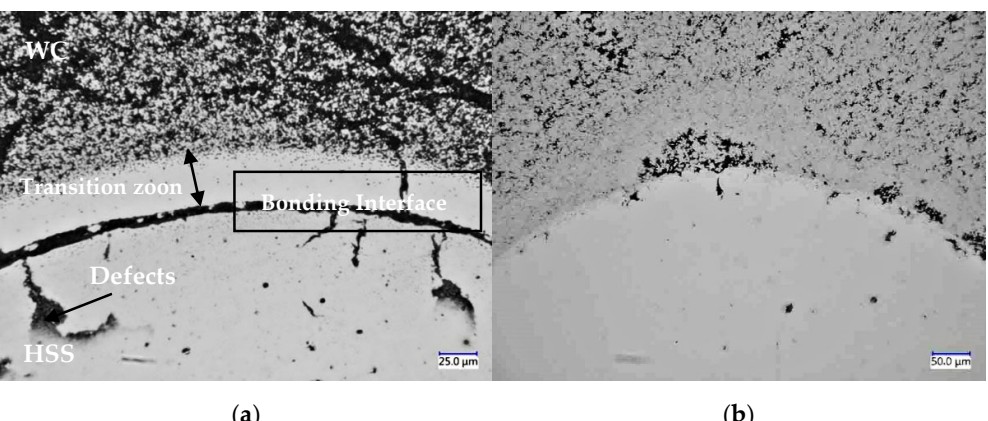

(**a**)  (**b**)

**Figure 4.** 1300 °C composite interface comparison (OM); (**a**) No Ni added (**b**) Ni added.

3.2.2. Influence on the Diffusion of Elements at the Interface of Composite Materials

The line scanning diagram of the WC/HSS composite interface prepared at 1300 °C for 90 min with and without added Ni is shown in Figure 5a,b. The dark area on the left side is the high-speed steel area and the light gray area on the right side is the WC area.

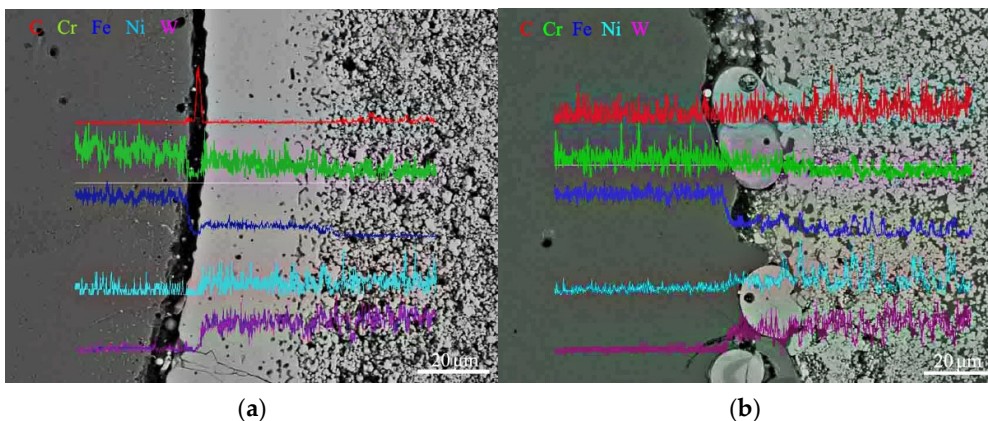

(**a**)                                                            (**b**)

**Figure 5.** Composite interface line scan with (**a**) no Ni added and (**b**) Ni added.

Figure 5a shows the interface of the WC/HSS composite material with no Ni added. The composite interface was cracked and a high-intensity peak corresponding to C at the crack can be observed, indicating that a large amount of C accumulated at the crack. Fe and Cr diffused from the high-speed steel region into the WC region. With the increase in the diffusion distance, the contents of Fe and Cr decreased. There was no obvious increase in W content or diffusion in the high-speed steel. Figure 5b shows the interface of the WC–Ni/HSS composite with Ni added. Fe and Cr diffused from the high-speed steel area into the WC area. Observing the changes in the element peaks, we found that Fe and Cr diffused into the WC area, mainly in the Ni bonding phase. The contents of Fe and Cr in the bonding phase area were much higher than in the WC structure. The diffusion of W was not obvious, which was potentially caused by the higher atomic mass of W. The addition of Ni to WC particles promoted the diffusion of Fe and Cr from the high-speed steel into the WC region, reduced the number of cracks at the interface and improved the interface bonding strength of the composite.

Figure 6a,b present an SEM micrograph of the bonding interface of the WC/HSS composite prepared at a sintering temperature of 1280 °C and with a holding time of 90 min with and without added Ni. The dark area on the right side of the figure is the high-speed steel area and the light gray area on the left is the WC area. Figure 6a depicts the SEM micrograph and the corresponding surface scanning (EDS) image at the interface of the WC/HSS composite with no Ni added. The figure shows that the banded zone between WC and high-speed steel was a transition zone with a width of approximately 19.74 μm. The microstructure of the transition zone was relatively dense, with fewer micropores. A large number of pores was observed in the WC area, resulting in low density. There were fine microcracks and long cracks located along the contact surface between the transition zone and the high-speed steel zone. Observing the element distribution diagram, the Fe content in the transition zone was higher than in the WC zone. The diffusion of W was not obvious. Figure 6b depicts the SEM micrograph and the corresponding surface scanning (EDS) image at the interface of the WC–Ni/HSS composite with Ni added. The figure shows that there was no obvious transition zone, and the WC structure near the interface between WC and high-speed steel was relatively dense. There were pores in the WC structures located far from the contact surface and pores in the high-speed steel region. There were microcracks on the contact surface of the composite material, but compared with the contact surface of WC–Co/HSS composite studied by Wang et al., the number of microcracks was obviously reduced, and no long cracks were found [8]. Observing the element distribution diagram, we found that the Fe distribution in the WC region was not uniform, and the content of Fe decreased with increasing distance from the high-speed steel. The diffusion of W was not obvious. The diffusion phenomenon of Fe and W

elements in the WC–Ni/HSS composites was extremely similar to that in the WC–Co/HSS composites studied by Wang et al. [8].

By adding Ni to WC particles, the contact surface defects of the composite materials were reduced, the diffusion of Fe elements was promoted, the W content per unit area in the WC region was reduced, the hardness of the WC structure was reduced, and the bonding strength between the reinforcement and the matrix of composite materials was improved.

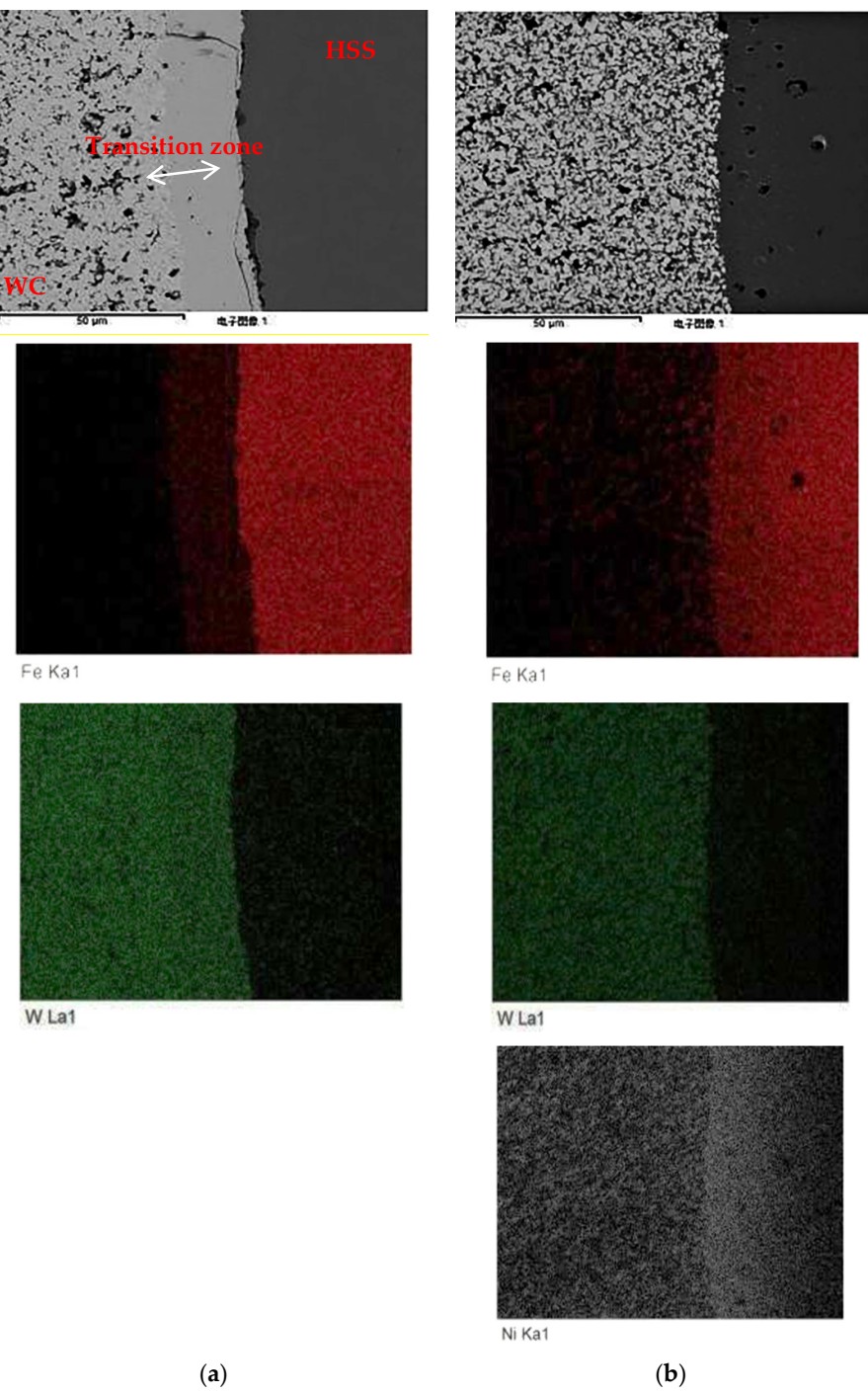

**Figure 6.** Composite interface scan of surface elements: (**a**) no Ni added; (**b**) Ni added.

### 3.3. Laser Confocal Experimental Results

Traditional detection methods can only observe the structure of the material under static conditions, and cannot observe the change in the structure during heating or cooling processes. High-temperature laser confocal microscopy (CLSM, Yonekura MFG Co., LTD, Osaka, Japan) can solve the above problems to a certain extent.

Figure 7a–c show the microstructural morphology of the WC–Ni/HSS composite samples with Ni added that were prepared at a sintering temperature of 1300 °C and held for 90 min under the laser confocal microscope during the heating process. In the figure, the left semicircle area is the high-speed steel area, and the surrounding region represents the WC–Ni structure. It can be seen that when the temperature reached 1202 °C (Figure 7a), the high-speed steel region began to melt near the composite interface. When the temperature reached 1216 °C (Figure 7b), the Fe in the high-speed steel began to diffuse into the WC–Ni region. When the temperature was 1250 °C (Figure 7c), the diffusion at the interface increased and more liquid phases appeared in the high-speed steel region. When the temperature was 1273 °C (Figure 7d), more WC precipitated at the grain boundary in the high-speed steel region. When the temperature was 1286 °C (Figure 7e), the grain boundary of high-speed steel began to melt. At 1350 °C (Figure 7f), the surface of the high-speed steel completely melted into the liquid phase. Since the WC–Ni structure in the composite material already had a high density during the first sintering, no obvious changes were observed during the reheating process using the laser confocal microscope.

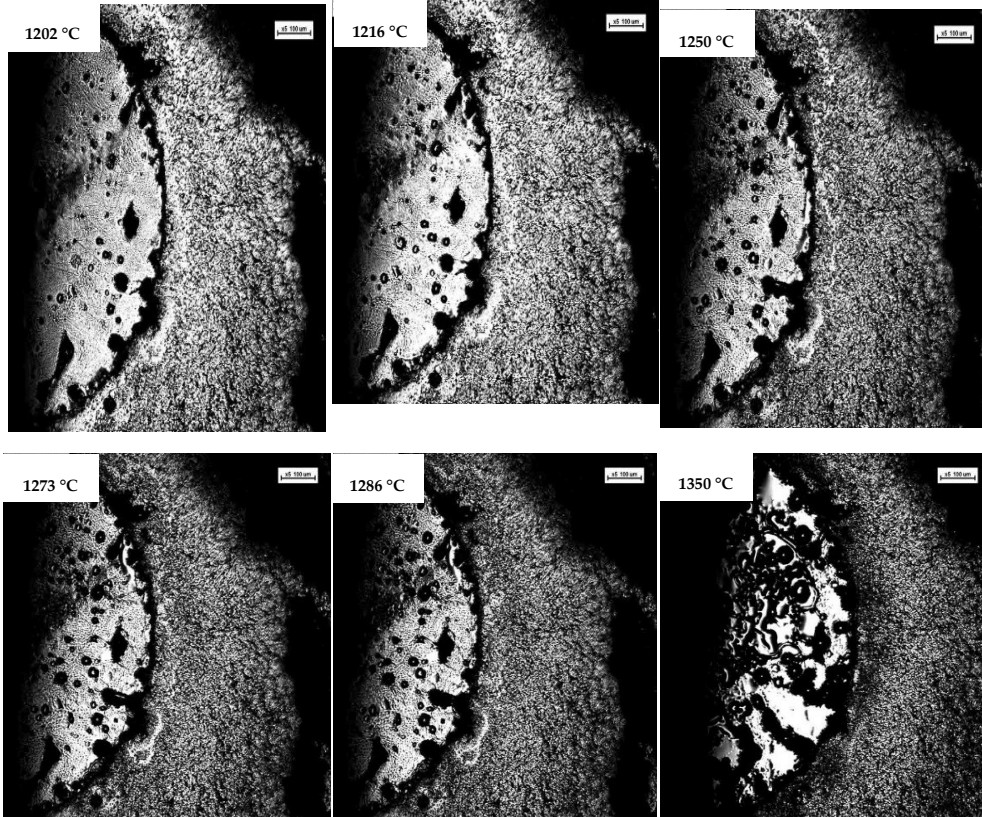

**Figure 7.** Composite interface changes of WC-Ni /HSS at ultra-high temperature laser confocal

Figure 8 shows the microstructural morphology of the WC/HSS composite samples without Ni that were prepared at a sintering temperature of 1300 °C and held for 90 min during the heating process under the laser confocal microscope. The left semicircle area is the high-speed steel area, while the surrounding area represents the WC structure. When

the temperature reached 1142 °C (Figure 8a), the high-speed steel near the composite interface defect began to melt. At 1184 °C (Figure 8b), the surface of the high-speed steel began to melt. At 1204 °C (Figure 8c), the grain boundaries of the high-speed steel began to melt. Then at 1220 °C (Figure 8d), a large area of liquid phase began to appear at the composite interface. As the temperature continued to rise, the more liquid phase formed on the surface of the high-speed steel, and the size of defects in the WC region gradually decreased. At 1378 °C (Figure 8f), the surface of the high-speed steel completely melted into the liquid phase. The size of the defects in the WC structure decreased.

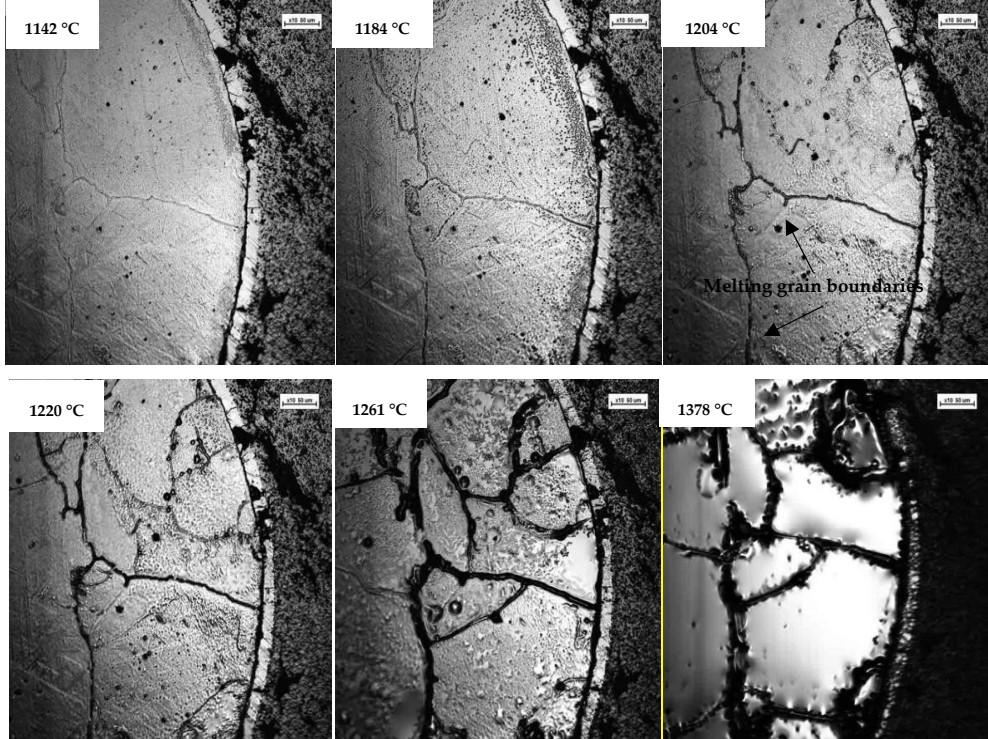

**Figure 8.** Composite interface changes of WC /HSS at ultra-high temperature laser confocal

### 3.4. Effects of Ni Addition on Microhardness of WC/HSS Composite

Figure 9a,b compares the hardness of the composite material at a sintering temperature of 1320 °C and holding time of 90 min, both with and without Ni added.

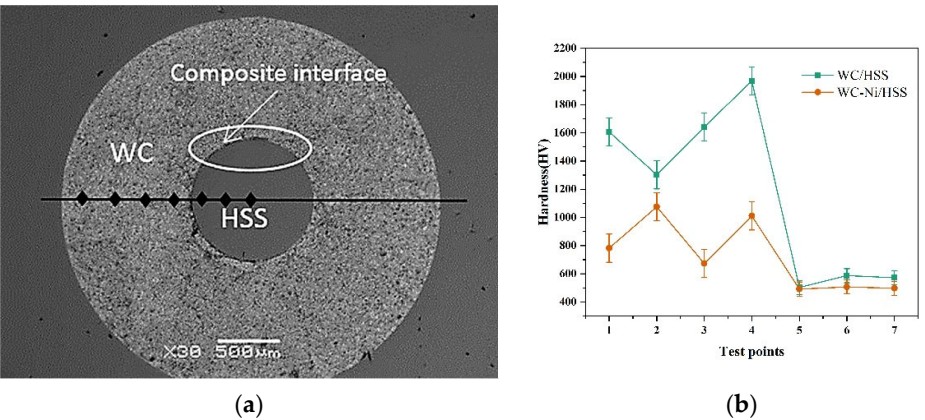

**Figure 9.** Comparison of the effect of adding Ni element on (**a**) dot position, and (**b**) hardness.

The highest value of hardness that was measured for WC without Ni was 1968 HV, which is higher than the maximum hardness of 1076 HV of WC with Ni. After adding Ni, the hardness of the material considerably decreased, which was in great agreement with the hardness results of WC with Co studied by Zhang et al. The microhardness of WC with Co just reached 1210 HV [23]. As Ni fills the pores between the WC structures, it reduces the content of the WC structure per unit area to a certain extent, as well as the effective load area of the indentation, and results in an increased pressure from the direct contact with the head per unit area of WC area, intensifying the plastic deformation in the direct contact area. Therefore, the size of the indentation increased. The addition of Ni resulted in the dissolution of more C during the high-temperature sintering process. When the temperature drops, the C precipitates and exists in the form of free C, which may also reduce the strength of the material.

From Figure 9b, it can be seen that the hardness near the bonding area of the composite material was relatively high. This is due to the diffusion of Fe in high-speed steel and the diffusion of Ni into the high-speed steel area, resulting in a decrease in the content of Ni in the diffusion area and the formation of iron–carbon compounds, which, in turn, led to a decrease in free C elements and an increase in hardness.

## 4. Conclusions

By comparing the microstructure, element diffusion, and microhardness of WC/HSS composites with and without the addition of Ni, we concluded the following:

(1) Ni fills the pores between the WC particles to increase the WC sintered density under high-temperature sintering. Ni prevents the agglomeration of WC grains, inhibits the abnormal growth of WC grains, and increases the uniformity of WC grain size;

(2) The addition of Ni promotes its diffusion between the composites, reduces defects such as pores and cracks, and improves the bonding strength between the composites;

(3) A relatively obvious diffusion phenomenon occurs from the high-speed steel region to the WC region at 1216 °C. When the temperature reaches 1350 °C, the surface of high-speed steel completely melts into the liquid phase;

(4) The addition of Ni can greatly reduce the hardness of the WC microstructure.

In general, the comprehensive properties of WC/HSS composites were improved by adding Ni. Whether from the perspective of cost-saving or resource-protection, this is of strategic significance to the micro-dill industry.

**Author Contributions:** experimental study, H.L. and L.M.; investigation, H.L.; writing—original draft preparation, H.L., L.M., and H.Z.; writing—review and editing, H.Z., Z.J. and L.Y.; project administration, Y.L. and F.A.; funding acquisition, L.Y. and H.Z. All authors have read and agreed to the published version of the manuscript.

**Funding:** This research was funded by the National the Natural Science Foundation of China (NSFC, Nos. 51474127 and 51671100) and the State Key Laboratory of Metal Material for Marine Equipment and Application-University of Science and Technology Liaoning Co-project (Nos. SKLMEA-USTLN 2017010 and 201905).

**Institutional Review Board Statement:** Not applicable.

**Informed Consent Statement:** Informed consent was obtained from all subjects involved in the study.

**Data Availability Statement:** Data sharing not applicable.

**Conflicts of Interest:** The authors declare no conflict of interest.

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
