# Peer review of "Effect of Ni on the Microstructure and Diffusion Behavior at the Interface of WC/High-Speed Steel Composites"

_metals, doi:10.3390/met11020341_

Round 1
Reviewer 1 Report
The paper has the potential to be interesting, but I have major concerns about similar papers produced by the same authors on the same topic. The authors must discuss how this study is different, and what new insights have been generated that deserve publication. Specific comments below:
Introduction:
Major correction: More background on WC-Co/steel bonding is needed. There are plenty of references on on this topic in the literature, including by the same authors, e.g. ref. [6]? How is this new? This must be described properly.
In line 26, authors mention “WC materials have been widely used for nuclear fusion devices”, but Refs. [1-3] do not mention nuclear fusion. Please give an appropriate reference, or references, for this statement.
It is not clear why the authors mentions refs. [17] to [19], which regard Co-based binders, not Ni or Steel, which is the subject of the paper. Please explain why the authors mention these references.
Lines 69-78: this discussion about effect of grain size on hardness and toughness is well known, and does not need such a long explanation. The authors can summarise in 1 sentence, with the appropriate reference.
Line 103, sentence is not clear. It sounds like the authors are using HSS powders in the binder, which is not the case. Suggest “matrix” should read “inner core” and “reinforcement” should read “outer sleeve” or similar.
Methods:
Line 113 please give details on how the WC+Ni powders were ball milled, e.g. solvents, milling container, milling time etc.
Lines 124: authors say “pressure of 10kN”. However kN is a unit for load, not pressure. Please give the pressure in MPa.
Results:
Line 141: authors say “density” was detected with image J. I think they mean porosity? Also, the authors must explain (in the methods) how Image J was used to calculate porosity. E.g. was there a thresholding applied? Or were the pores traced manually?
Major correction: The porosity measurements need to be supported with further information. The authors should be able to estimate the density from the height of the as-pressed- and as-sintered specimens, their weight, and the weight of the original steel die. Please also give the density both before and after sintering. How do the results from specimen size and Image-J compare? What is any difference caused by? This may need explaining.
Fig. 3(a). The authors claim to have measured an average grain size of 1.57 um, however the image does not show clear grain boundaries, due to the lack of binder. How are grain boundaries quantified to such accuracy? I suspect the grain size is much lower that 1.57 um, since the lack of binder should decrease the rate of diffusion, and by extension, grain growth. I suggest the author either provide better quality images, or otherwise delete the grain size measurements for pure WC, as they may be inaccurate.
Line 195, the authors mention sintering temperature as the origin of the failure. However, Fig. 8 shows that the cracking forms even at low temperature (e.g. 1142 C) Is it not important to mention thermal expansion mis-match? What is the difference for WC and HSS?
Figure 5: the figure must be improved:
- the EDX results are very noisy. Please can the authors use a smoothing function to make the data clearer?
- Also, the element labels are not visible. Please put labels on a white background and increase text size.
- There is no scale-bar. Please provide one that is clear.
LINE 223: authors say: “Ni promotes more diffusion of Fe”. However, there is a great-deal of diffusion in the pure WC case, as evidenced from the large, pore-free zone between the crack and the porous WC (labelled “binding interface” in Fig 4a, and labelled “transition zone in Fig. 6a). Surely this zone is evidence of mass FeCr migration? Please be consistent with terminology as well.
Authors mention transition zone is 19.7 um at 1280 C, what is the zone width at 1300 C? It looks larger to me. Please discuss.
Line 258-261. These are methodological statements. Please put this information in the methods section.
Please also describe the confocal microscopy set-up in more detail within the methods. Where is the microscope in relation to the sintering furnace (perhaps a diagram can be added in Fig. 1)? How is the temperature monitored? What is the sintering atmosphere? Please give details, as this kind of experiment is not standard within the scientific community.
Figure 7: this needs improvement:
- please increase the size of the text so that the scalebar is legible
- it would help if the temperature was put in the top-left, rather than in the caption.
- Please use arrows and labels to indicate relevant features.
Figure 8, what are the linear features in the steel core? Are these cracks? Please explain.
Figure 9b, hardness plot is not clear. The image quality and size of text need increasing. Error bars are needed.
The hardness measurements should be explained in the methods. What was the load, how many indents were made at each position?
Major correction: Please comment on the results in light of similar studies in the literature, e.g. WC-Co bonded to steel. Are the results similar or different? Please comment on any differences in behaviour.
Reviewer 2 Report
The authors are to be congratulated for conducting a fine piece of work on the Effect of Ni on the Microstructure and Diffusion Behavior at the
Interface of WC/High-Speed Steel Composites and describing it so clearly. The English and writing is virtually perfect, but there was no way to say this in the review questions.The claims by the authors are well supported by the presented results. This reviewer has only two suggestions to make:
1. The arrows shown in Fig. 3(b) do not seem to clearly indicate which phase is which. Perhaps the arrows can be more carefully drawn or the authors can tell us which is the dark phase and which is the light phase.
2. In Fig. 5 the symbols identifying the different element traces are hard to read. Putting easy-to-read labels at each trace would be better.
Otherwise, a fine job.
Reviewer 3 Report
There is a group in Portugal around Mrs. Senos and Fernandez, who is quite active in research about steel binders in cemented carbides. The published work of this research group has completely been neglected in the paper. It is necessary that the authors provide a actual view of references and literature on this topic when they publish this study.
Round 2
Reviewer 1 Report
Please see the attached response

Reviewer 3 Report
I can accept the amended manuscript although there is much more important work of Fernandez, Senos et al., which deserved to be regarded and compared to the results
Author Response
I am very grateful to your comments for the manuscript. After carefully studying the comments and your advice, we have revised and edited the paper in detail.
- I can accept the amended manuscript although there is much more important work of Fernandez, Senos et al., which deserved to be regarded and compared to the results.
Response: Thanks for your comments on our paper. We have looked up the data again. The articles published by these two authors are indeed worthy of attention. We also explain their achievements in this paper, adding references 12 and 13 (Lines47-51).
I hope you are satisfied with the revised version, however, if there is more question, we are willing to revise it again.
Thank you.